# Effect of Acid Leaching Pre-Treatment on Gold Extraction from Printed Circuit Boards of Spent Mobile Phones

**DOI:** 10.3390/ma14020362

**Published:** 2021-01-13

**Authors:** Nicolò Maria Ippolito, Franco Medici, Loris Pietrelli, Luigi Piga

**Affiliations:** 1Department of Industrial and Information Engineering and Economics, University of L’Aquila, Via Giovanni Gronchi 18, Zona industrial Pile, 67100 L’Aquila, Italy; nicolomaria.ippolito@univaq.it; 2Department of Chemical Engineering, Materials and Environment, Sapienza University of Rome, Via Eudossiana 18, 00184 Rome, Italy; luigi.piga@uniroma1.it; 3Department of Chemistry, Sapienza University of Rome, P.le Aldo Moro 5, 00185 Rome, Italy; lpietrelli@gmail.com

**Keywords:** printed circuit boards, spent mobile phones, thiourea, precious metals, hydrometallurgy, factorial plans

## Abstract

The effect of a preliminary acid leaching for the recovery of gold by thiourea from printed circuit boards (PCBs) of spent mobile phones, was investigated. Preliminary leaching is aimed to recover copper in the leachate that would compete with gold in the successive leaching of the residue with thiourea, thus preventing the formation of the gold-thiourea complex. Two hydrometallurgical routes were tested for the recovery of copper first, and gold after. The first one was based on a two-step leaching that utilizes sulfuric acid and hydrogen peroxide in the preliminary leaching and then thiourea for the recovery of gold in the successive leaching: A copper and gold recovery of 81% and 79% were obtained, respectively. In the second route, nitric acid was used: 100% of copper was recovered in the leachate and 85% of gold in the thiourea successive leaching. The main operative parameters, namely thiourea and ferric sulphate concentrations, leach time, liquid-solid ratio, and temperature were studied according to a factorial plan strategy. A flowsheet of the processes was proposed, and a mass balance of both routes was obtained. Finally, qualitative considerations on the technical and economic feasibility of the different routes were made.

## 1. Introduction

Electrical and Electronic Equipment (EEE) were significantly growing all over the world before the corona virus crisis, owing to the continuous development of the technology. Therefore, the amount of waste of this equipment (WEEE) was expected to increase in the next few years [1]. Spent mobile phones belong to WEEE. Consumers are driven to change their mobile phone by the continuous production of new and fashionable models with higher functionality so that the average usage of these components is around 2 years.

The presence of recoverable gold, in printed circuit boards (PCBs) of spent mobile phones, as well as of silver and palladium, makes them among the most valuable components of WEEE, and their recycling would also reduce the environmental impact of the waste due to the presence of heavy metals like copper and nickel [2]. Moreover, the recovery of gold would improve the collection, treatment and recycling of spent mobile phones then contributing to the circular economy according to the EU directive [3]. PCBs of mobile phones are made of 63% metals, 24% ceramics and 13% polymers [4], copper being the main component among the metals. The concentration of gold is higher than the one present in typical ores [5,6,7,8] and has changed as a function of the different types and of the year of production of mobile phones; the tendency is to reduce gold in the PCB to lower their production costs. Li et al. [9] determined the concentration of metals contained in PCBs mobile phones: Cu 39.9%, Ag 540 mg/kg and Au 43 mg/kg. Camelino et al. [10] detected Cu 65.7%, Ag 285 mg/kg, Au 168 mg/kg, and Pd 110 mg/kg. Petter et al. [11] analysed the composition of mobile phones from different companies and year of manufacture and found that the highest concentration of Au was 880 mg/kg. Despite the lower content of gold in PCBs with respect to the content of copper, its higher value makes that 90% of the economic potential of the PCBs phones is given by gold.

A number of hydro-metallurgical, pyro-metallurgical [12] and bio-metallurgical [13,14,15] processes have been proposed for the recovery of gold [16]. Most processes use hydrometallurgical techniques for economic and environmental reasons that provide the dissolution of most metals as a preliminary step and then the dissolution of gold in alkaline or acid media. The process generally starts with a mechanical treatment that includes the shredding in order to increase the liberation grade of valuable metals. A thorough literature survey on the current status of leaching of precious metals was published 8 years ago by Zhang et al. [17]. Li et al. [9] studied the dissolution of gold and silver in thiourea solution taking into account the influence of particle size, thiourea and oxidant (Fe^3+^) concentrations and temperature. The best conditions that permitted to obtain 90% of gold dissolution and 50% of silver dissolution were: particle size 150 µm, thiourea concentration 25 g/L, Fe^3+^ = 0.6% *w*/*v*, room temperature, and 2 h of leach time. Gurung et al. [18] found that the best conditions to dissolve 3200 mg/kg gold present in PCBs were 38 g/L of thiourea, 0.05 mol/L sulfuric acid at 45 °C and 53–75 µm particle size. Copper is present in PCBs in a concentration of around 30% and with the formation of the complex Cu-thiourea enhances the consumption of thiourea that was expected to form the complex gold-thiourea that is the necessary step to achieve the dissolution of the precious metal [10,19]. Birloaga et al. [20] reports the leaching of spent computer-printed circuit boards, which are different from the boards of spent mobile phones that are the material used in this paper.

Considering the scarcity of publications that deeply illustrate the influence of copper removal before the gold leaching with thiourea from PCBs spent mobile phones, the aim of this work was to study the effect of a preliminary treatment with H_2_O_2_/H_2_SO_4_ or HNO_3_ to highlight the differences occurring on the successive gold extraction when the two different oxidant solutions are preliminarily applied on the same material. In order to optimize the whole process, experimental tests were carried out with the use of a few statistical factorial plans to investigate the effect of the following factors: thiourea and ferric sulphate concentration, leach time, liquid-solid ratio, and temperature. This also permitted to determine the influence of those experimental parameters on the gold extraction against the purely experimental error. Therefore, a qualitative technical-economic comparison between the two preliminary treatments was proposed. The experimental data were used to describe two possible alternative flowsheets and their respective mass balances according to the best operative conditions.

## 2. Fundamentals

Gold is a noble metal and is not oxidized neither in water nor in air. The dissolution of gold according to the reaction (1):Au^+^ + e^−^ ⥦ Au^0^(1)
does not practically occur due to the high standard Gibbs free energy of formation of the ion Au^+^ (ΔG_f_^0^ = 163,063 kJ/mol). The standard redox potential, E^0^, of this reaction is 1.69 V, and is easily calculated by Equation (2):ΔG_r_^0^ = −nFE^0^(2)
where ΔG_r_^0^ = −ΔG_f_^0^, n is the number of electrons involved in the reaction and F is the Faraday constant 96,487 C/mol. So, gold cannot be oxidized and dissolved even by concentrated nitric acid (NO_3_^−^ + 4H^+^ + 3e^−^ → NO + 2H_2_O, E^0^ = 0.96 V). However, the standard potential is related to a unitary activity (a = 1) and not to the effective activity of gold ions in the solution. Hence, the effective redox potential of reaction (1) is calculated by Nerst Equation (3):E = 1.69 + 0.059 log [Au^+^] at 25 °C,(3)

The reaction between the ion Au^+^ and particular ligands forms very stable complexes, according to the following reaction (4):Au^+^ + nL^y−^ ⥦ AuLn^1−ny^(4)
and leads to a decrease of the activity of Au^+^ ions in the solution, which are blocked in the complex, and the effective redox potential of reaction (1) decreases. Usually, CN^−^ is used as a ligand in gold treatment that is an environmentally undesirable reagent but thiourea (CS(NH_2_)_2_) can replace CN^−^ even though thiourea is in suspicion of being a carcinogen for long exposures. If thiourea is added to the solution where solid gold is present in equilibrium with its Au^+^ ions, the following reaction can be written, which spontaneously proceeds towards the formation of the Au-thiourea complex [21]:Au^+^ + 2 CS(NH_2_)_2_ ⥦ Au(CS(NH_2_)_2_)_2_^+^(5)

The stability constant of the Au-thiourea complex can be expressed as:(6)$$K_{\mathrm{st}}=2\times 10^{23}=\frac{{\left[\mathrm{Au}{\left(\mathrm{CS}{\left({\mathrm{NH}}_{2}\right)}_{2}\right)}_{2}\right]}^{+}}{[\mathrm{Au}{]}^{+}[\mathrm{CS}{\left({\mathrm{NH}}_{2}\right)}_{2}{]}^{2}}\;$$

If reaction (1) is subtracted from reaction (5), the following reaction is obtained
Au(CS(NH_2_)_2_)_2_^+^ + e^−^ ⥦ Au^0^ + 2 CS(NH_2_)_2_(7)
whose effective reduction potential is calculated by substituting [Au^+^] of Equation (6) in Equation (3):(8)$$E\;=\;0.315\;+\;0.059\;\log\;\frac{[\mathrm{Au}(\mathrm{CS}{\left({\mathrm{NH}}_{2}{)}_{2}{)}_{2}\right]}^{+}}{{\left[\mathrm{CS}{\left({\mathrm{NH}}_{2}\right)}_{2}\right]}^{2}}$$

This means that the system Au^+^/Au^0^, in the presence of thiourea, becomes reduced with respect to other systems like oxygen for example, (2 H_2_O ⥦ 4 H^+^ + O_2_ + 4 e^−^, E^0^ = 1.23 V). Nevertheless, the fugacity of oxygen in a water solution is not sufficient to achieve the proper potential to oxidize gold. On the contrary, the lower potential value of the reaction that forms the Au-cyanide complex (E^0^ = −0.57 V) at alkaline pHs, allows the gold to be oxidized by the oxygen dissolved in water. Moreover, the reduction of oxygen is slow in thiourea solutions and a stronger oxidizing agent is needed. Very strong oxidants should be avoided when working with thiourea that is oxidized to formamidine disulphide, which causes loss of thiourea and then decomposes irreversibly into cyanamide and sulphur [22]. Sulphur can passivate the surface of gold and prevent it from being dissolved. Moreover, precipitation of sulphur can drag a part of the Au-thiourea complex, then decreasing the recovery of dissolved gold in the solution. In practice, a medium oxidant has to be used to dissolve Au, like Fe_2_(SO_4_)_3_, being 0.77 V the standard potential of the equilibrium reaction Fe^3+^ ⥦ Fe^2+^ [23]. This oxidant was used in the leaching tests shown in this paper. The total gold dissolution is shown by reaction (9):2 Au^0^ + 4 CS(NH_2_)_2_ + Fe_2_(SO_4_)_3_ ⥦ [Au(CS(NH_2_)_2_)_2_]_2_ SO_4_ + 2 FeSO_4_(9)

The potentials of the electrochemical reactions cited in the text, as a function of pH, are reported in Figure 1, and show a potential-pH diagram that was constructed by the authors taking into account the relations between the equilibrium constants of those reactions and their standard Gibbs free energy, whose numerical values at 298 K were taken by Gaspar et al. [24]. The potentials reported in the figure were calculated with the actual concentration of gold present in the PCBs used in this study and the concentrations of thiourea and ferric sulphate used for the experimental tests, that were 5×10^−4^ mol/L (all gold dissolved), 0.33 mol/L and 0.11 mol/L (expressed as Fe^3+^ ), respectively.

The equilibrium concentration of the species involved can be calculated as reported in Table 1.

Considering that X can be up to 5 × 10^−4^ mol/L, the thiourea concentration at the equilibrium may be approximated by 0.33 mol/L. After simple calculations, the concentrations of Au^+^ and of the Au-thiourea complex at the equilibrium are 2.3 × 10^−26^ mol/L and 5 × 10^−4^ mol/L, respectively. The redox potential of reaction 7 decreases up to 0.18 V and is shown up to pH 4 as the stability of the complex decreases at higher pHs, depending on both the pH and on the gold and thiourea concentrations [24]. The redox potential of the equilibrium reaction Fe^3+^ ⥦ Fe^2+^ is 0.71 V and is able to oxidize and dissolve gold according to reaction (9).

## 3. Materials and Methods

### 3.1. Preparation and Characterization of Waste PCBs

Experimental studies were carried out with 10 kg of mobile phone PCBs supplied by a WEEE recycling plant that had previously removed all the electronic components like capacitors, cables, resistors, etc. The sample was firstly crushed by a Retsch SM 2000 cutting (Haan, Germany) mill up to −4 mm after various steps of comminution, reducing the size of the output grid after each step. This product was further ground with a Fritsch pulverisette 9 vibratory steel ring mill for 10 min. The resulting powder was sieved with a 0.5 mm screen and the +0.5 mm and −0.5 mm fractions were obtained. The +0.5 mm fraction was analysed with X-ray fluorescence by using a Bruker-Tracer IV SD (Billerica, MA, USA). The −0.5 mm fraction was subjected to an automatic sampling with PT 100—Retsch to obtain representative samples for chemical analyses and leaching tests. Chemical analyses of this fraction were carried out on six 1 g portions of the powder that were dissolved with 1:3 nitric acid and hydrochloric acid (aqua regia) at 90 °C for 3 h. The cooled digestion solution was filtered to remove plastics and ceramics and analysed for the main constituents and precious metals by flame atomic absorption spectroscopy (AAS).

### 3.2. Leaching Experiments and Test Planning

All chemical reagents utilized for leaching experiments were analytical grade: H_2_SO_4_ (96%), H_2_O_2_ (30% *w*/*v*), HNO_3_ (65%), CS(NH_2_)_2_ (99%), and anhydrous Fe_2_(SO4)_3_. The leaching tests were performed in a 250 mL bottle flask with 10 g of sample powder and 100 mL of leaching solution under 250 rpm stirring and constant pH 1. Samples of the leaching solution were taken at different times and analysed by AAS for copper and gold, in order to study the kinetics of extraction. At the end of each test, the solution was separated from the residue by centrifugation; no paper filter was used to avoid loss of gold in the filter. The residue was washed, dried at 105 °C, weighed, and then attacked with aqua regia to determine the content of copper and gold not dissolved that was added to the content of the two metals in the solution. This procedure was used due to the heterogeneity of powders of PCBs and permitted to determine the reconstituted feed that was taken as the concentrations of the metals in the initial sample to calculate the recovery of metals.

Firstly, leaching tests were carried out to evaluate gold dissolution directly by thiourea solution [9], with no preliminary treatment to remove copper that competes with gold for the formation of their respective complexes with thiourea. These tests were carried out on different particle-size fractions of the milled PCBs. After this, the two preliminary leaching treatments were studied to dissolve selectively copper from PCBs, before the successive thiourea leaching to recover gold left in the residue. These treatments can be represented by reactions (10) and (11).
(10)$$\mathrm{Cu}+{\;H}_{2}{\mathrm{SO}}_{4}+H_{2}O_{2}\to {\;\mathrm{CuSO}}_{4}+2{\;H}_{2}O$$
(11)$$\mathrm{Cu}+\;4{\mathrm{HNO}}_{3}\to \;\mathrm{Cu}\;{\left({\mathrm{NO}}_{3}\right)}_{2}+2{\;\mathrm{NO}}_{2}+2{\;H}_{2}O$$

The H_2_O_2_/H_2_SO_4_ treatments [25] were carried out in two consecutive steps. In the first step, the −0.5-mm fraction was leached with a relatively low sulphuric acid concentration. The residue of leaching was vacuum filtered, washed and then leached with a higher concentration of sulfuric acid in the second step. Three treatments were carried out, the first with 1 mol/L and 2 mol/L, the second with 2 mol/L and 3 mol/L and the third with 3 mol/L and 4 mol/L sulfuric acid. In the HNO_3_ treatment, the copper extraction from the −0.5 mm fraction, as a function of leach time and of the acid concentration in the range 0.1–6 mol/L, was investigated.

On the basis of the experimental data, the residues obtained by preliminary acid treatments were leached with thiourea and ferric sulphate to evaluate how the two preliminary treatments affected the gold dissolution.

The thiourea leaching tests on the residues of the two preliminary tests were planned according to a 2^3^ full factorial design whose three investigated factors and two levels adopted (−1 and +1) are reported in Table 2. Three replicated tests were carried out to evaluate the purely experimental error at a middle level (0).

These operating parameters were kept constant during the tests: liquid-solid ratio (L/S) 10, temperature (T) 25 °C, H_2_SO_4_ 0.1 mol/L (pH 1), stirring 250 rpm, and leach time 1 h. The results were elaborated by analysis of variance [26]. Finally, on the basis of the results obtained by factorial experimentation, a further full factorial plan (2^2^) was studied, with the aim to evaluate the effect of thiourea at higher concentration and of the L/S ratio (Table 3). Duplicated tests were carried out at middle level.

## 4. Results

### 4.1. Characterization of Waste PCBs Mobile Phones

The +0.5-mm fraction of the milled PCBs was 11.5% of the initial material. Visual inspection revealed the presence of metal lamellae not further grindable. XRF analysis showed that the main elements were Fe 51.4%, Cu 15.0% and Cr 13.7%, while no gold was detected.

Chemical composition of the −0.5 mm fraction is reported in Table 4.

The main component is Cu (27.0 ± 1.2%) followed by the major elements (Si, Fe, Sn Al). Regarding the precious metals, the following concentrations were detected: Au 439 mg/kg, Ag 336 mg/kg, and Pd 18 mg/kg. Similar values were reported by Ning et al. [27] and by Pietrelli et al. [7]. Pd concentration was low because the palladium-rich components were previously removed. Heavy metals were also present, like Ni, Zn and Cr in concentrations of 14,500 mg/kg, 5900 mg/kg and 1300 mg/kg, respectively, and it would be necessary to take account of this for the safe disposal of process residues.

The low standard deviations indicate the good homogeneity of the −0.5 mm fractions involved in characterization and in further leaching tests.

### 4.2. Thiourea Leaching with No Preliminary Treatment

Thiourea leaching tests were firstly carried out under the following operative conditions: thiourea 25 g/L, Fe^3+^ = 0.6% *w*/*v*, H_2_SO_4_ 0.1 mol/L (pH 1), L/S ratio 10, T 25 °C, and stirring 250 rpm [9,18,28]. The effect of leach time (0.5, 1, 2 and 3 h) and of the size of PCBs (full board, half board, particle size of 1 cm, 1 mm and 0.5 mm) on gold dissolution was studied.

The results reported in Figure 2 show that gold extraction was negligible in any condition and did not reach 12%, contrary to what was reported by Li et al. [9].

The leach time and the size of PCB slightly affects the dissolution of gold. Between 0.5 h and 1 h of leach time, no consistent difference among dissolutions was observed and the effect of particle size seems to be irrelevant at size larger than 1 cm, while the higher gold dissolutions occurred for sizes smaller than 1 mm, due to the higher degree of liberation of the gold-containing particles. There is a decrease of gold dissolution with leaching times higher than 1 h and, after the same time, the solution turned from clear to turbid, probably owing to the degradation in acidic solution of thiourea into formamidine disulphide that produces elemental sulphur and cyanamide according to reaction (12) [29].
(12)$$\mathrm{CS}({\mathrm{NH}}_{2}{)}_{2}\;+\;2{\mathrm{Fe}}^{3+}\;\to \;{\mathrm{NH}}_{2}\mathrm{CN}\;+\;2{\mathrm{Fe}}^{2+}\;+\;S\;+\;2H^{+}$$

Therefore, it was assumed that leach times higher than 1 h are not suitable to maximize gold dissolution. On the basis of these tests, gold dissolution was evaluated on PCB particles of size −0.5 mm at higher concentrations of thiourea and Fe^3+^. After 1 h, 10% of gold dissolution was achieved with 50 g/L thiourea and 1.2% *w*/*v* Fe^3+^, while 15% was achieved with 130 g/L thiourea (close to the solubility) and 3.1% *w*/*v* Fe^3+^.

Low gold dissolutions are due to the high content of copper (around 30%) in PCBs, which has a strong negative effect on the kinetics of the gold-leaching reaction, because copper competes with gold for thiourea with the formation of Cu[CS(NH_2_)_2_]_2_^2+^ that prevails over the formation of Au[CS(NH_2_)_2_]_2_^+^. This occurrence is less pronounced when gold is recovered from primary raw materials due to the negligible presence of copper in the siliceous or sulphide matrix where gold is generally present. For this reason, preliminary treatments with sulfuric or nitric acid were studied to firstly extract copper and ensure that the thiourea used for the subsequent leaching was mainly used to dissolve gold.

### 4.3. Sulfuric Acid and Hydrogen Peroxide Preliminary Treatment

In Figure 3, copper dissolution yields as a function of selected sulfuric acid concentration for each leaching stage are shown.

At the lowest sulfuric acid concentrations (1 and 2 mol/L), 28% of copper was dissolved in the first step and 33% in the second step for a total of 61%. At higher sulfuric acid concentrations (2 and 3 mol/L), most of the copper was dissolved in the first step (55%) and 26% in the second step for a total of 81%. Slightly better results were achieved with the highest concentration investigated (3 and 4 mol/L). Each copper extraction value was the average of three values, as each treatment was carried out three times. The other operative parameters were kept constant, namely H_2_O_2_ 20% v/v, L/S ratio 10, T 25 °C, leach time 1 h, particle size −0.5 mm, and stirring 250 rpm. The recovery of copper as a metal can then be performed by electrodeposition directly from the sulphuric solution [30]. The residue of previous leaching was leached with thiourea with the same operative conditions as those used for the leaching carried on PCBs of size −0.5 mm without preliminary treatment: thiourea concentration 25 g/L, Fe^3+^ = 0.6% *w*/*v*, H_2_SO_4_ 0.1 mol/L (pH 1), L/S ratio 10, T 25 °C, and leach time 2 h, under stirring. The results of thiourea dissolution with the preliminary treatment are reported in Table 5, where the results without preliminary treatment are also shown.

After 0.5 h leach time, the 79% of gold was dissolved and no further increase was achieved after 1 h leaching, while a slight decrease in dissolution was obtained after 2 h. This behaviour was probably tied to thiourea degradation as already discussed. These values were determined as an average of three replicated tests with low standard deviations (<1.4%).

### 4.4. Nitric Acid Preliminary Treatment

In Figure 4, the results of nitric acid tests are shown. For each test, the following operative conditions were kept constant, L/S ratio 10, T 25 °C. Tests were carried out with three replications, experimental error was evaluated to be in the range of 1–2%. All the copper was extracted with 3 mol/L acid concentration, after 1 h leach time.

The leach time needed to achieve total dissolution of copper increased to 2 h when the acid concentration was lowered to 2 mol/L. Below 2 mol/L acid concentration, regardless of the leach time, no satisfactory extraction of copper was achieved. Therefore, on the basis of these results, only the residue of previous 3 mol/L HNO_3_ treatment was leached with thiourea at the same operative conditions that were applied to the solid residues obtained from H_2_O_2_/H_2_SO_4_ treatment. Gold dissolution of 81% and 80% after 1 h and 2 h leach time, respectively, were achieved. Even though the HNO_3_ treatment allowed to remove 100% of the copper present in PCBs, the increase of the gold dissolution was not significant with respect to the preliminary treatment with H_2_O_2_/H_2_SO_4_ that was able to dissolve as much as 81% of copper.

### 4.5. Influence of the Main Operative Variables on Dissolution of Gold

The influence of thiourea concentration (factor A), Fe^3+^ concentration (factor B) and temperature (factor C), was studied according to a 2^3^ full factorial plan that is shown in Table 6 with the levels assigned to each factor and the gold dissolution yields.

Three replicated tests were carried out in the middle of the plan to determine the purely experimental error. The highest dissolution range 78.4–81.8% was achieved with the treatments “a”, “ac” and “abc”, at the highest level of thiourea concentration (factor A, 40 g/L). The lowest dissolution range, 68.2–71.9%, was achieved at the lowest level of thiourea concentration (factor A, 10 g/L). Factors B and C seem not to affect gold dissolution at 95% confidence level. This is confirmed by the results of analysis of variance reported in Table 7.

There is a significant positive effect of thiourea concentration (factor A) on dissolution of gold while factors B, C, and all the interactions among the factors are not significant. Based on the results obtained by the plan, the following relation was obtained:Au dissolution (%) = 75.65 + 4.70 × thiourea concentration (g/L)

Finally, a further full factorial plan (2^2^) was studied in order to investigate the effect of higher thiourea concentrations on gold dissolution while the other already studied factors (Fe^3+^ concentration and temperature) were kept at a low level since they were not significant. The results are reported in Table 8.

The maximum gold extraction (85.2 ± 0.8%) was achieved at central point tests with the following conditions: thiourea 60 g/L (factor A), L/S ratio 10 (factor B). The analysis of the data, according to Yate’s algorithm, showed that by the F-test method at 95% confidence level, factor A and factor B are not significant within the investigated value range.

## 5. Discussion

### 5.1. Gold Dissolution Treatments Comparison

A summary of the results obtained by the investigated treatments and a qualitative technical and economic feasibility are reported in Table 9.

Recoveries of gold and copper are referred to thiourea leaching tests carried out with the same operative conditions. Direct thiourea leaching is not suitable to dissolve gold because of the high amount of copper in phone PCBs. Both preliminary treatments allow similar gold recovery in the successive thiourea leaching, but the treatment with nitric acid permits the total recovery of copper. Despite different copper recoveries being obtained by the two routes, gold dissolutions after thiourea leaching were similar, probably because after H_2_SO_4_/H_2_O_2_ preliminary treatment, non-leached copper is present as a refractory material and did not compete with gold for thiourea consumption. Moreover, the higher recovery of copper affects only 1% on the revenue of valuable metals, with respect to the sulfuric acid–hydrogen peroxide treatment.

The comparison between the two preliminary treatments must be made in terms of plant complexity, process management and disposal of waste. The use of hydrogen peroxide must be considered in terms of chemicals increase, complexity (storage) and, therefore, costs. The pre-treatment with HNO_3_ will produce a large amount of NOx and exhausted NaOH and Ca(OH)_2_ resulting from the treatment of off-gas. Despite nitric acid being less expensive than the solution sulfuric acid–hydrogen peroxide, the recovery and the grade of copper by electrodeposition from sulfuric acid solutions are higher than those obtained from nitric acid solutions [29].

### 5.2. Mass Balances

On the basis of experimental results, two flowsheets for the processing of 1 kg of PCBs from end-of-life mobile phones were described. The first scheme is reported in Figure 5.

The flowsheet includes: First, leaching with sulfuric acid and hydrogen peroxide, filtration, drying of solid residue; and second, leaching with sulfuric acid and hydrogen peroxide, filtration and washing, drying of solid residue, leaching with thiourea, and centrifugation and drying of the solid residue. According to this process, the following products are obtained: two leach solutions enriched in copper (148 g of copper in the first and 70 g of copper in the second are dissolved) and a leach solution in which 347 mg of gold are dissolved. Copper and gold can be recovered by electrodeposition [31].

The second flowsheet is reported in Figure 6.

The flowsheet includes: the use of nitric acid leaching, filtration and washing, drying of solid residue, thiourea leaching, and centrifugation and drying of solid residue. The solid residue of both processes is mainly constituted by plastics and ceramics to dispose of or to be used as filler to reinforce the new polymer plastic items such as coating and carpets. The residual solutions can be reused, with the necessary make-up of chemicals, for a few cycles [32]; after that, the exhausted solutions can be sent to chemical-physical operations for abatement of pollutants and successive recycling or disposal.

## 6. Conclusions

Printed circuit boards of spent mobiles contain 27% of Cu as the main component and precious metals like Au 439 mg/kg, Ag 336 mg/kg and Pd 18 mg/kg. In spite of the low content of Au with respect to copper, 90% of the value of board is given by gold. The recovery of gold from the board “as is” with thiourea leaching was not feasible due to the low liberation grade of the metal, and a recovery of only 15% was obtained. Direct leaching with thiourea on the board comminuted to 0.5 mm, and failed owing to the competition between copper and gold for thiourea to form the respective complexes. Therefore, two pre-treatments to solubilize copper and to facilitate the subsequent gold dissolution were tested. The first treatment with sulfuric acid and hydrogen peroxide led to the recovery of 81% of copper in the sulfuric solution after 0.5 h leaching and of 79% of gold in the successive thiourea leaching. The second pre-treatment with 3 mol/L nitric acid achieved dissolution of the totality of copper after 1 h leaching, and 81% of gold was then recovered in the successive thiourea leaching. Analysis of variance applied on the results of the experimental tests highlighted that only the concentration of thiourea is significative; ferric sulphate concentration, leach time, liquid-solid ratio, and temperature seem not to affect the recovery of gold, at least in the range of values investigated. In fact, an increase of thiourea concentration allowed to achieve 85% gold recovery. 

Finally, taking into account the mass balance of copper and gold, the technical and economic feasibility and the lower environmental impact, the preliminary treatment with sulfuric acid and hydrogen peroxide seems to be the most promising process.

## Figures and Tables

**Figure 1 materials-14-00362-f001:**
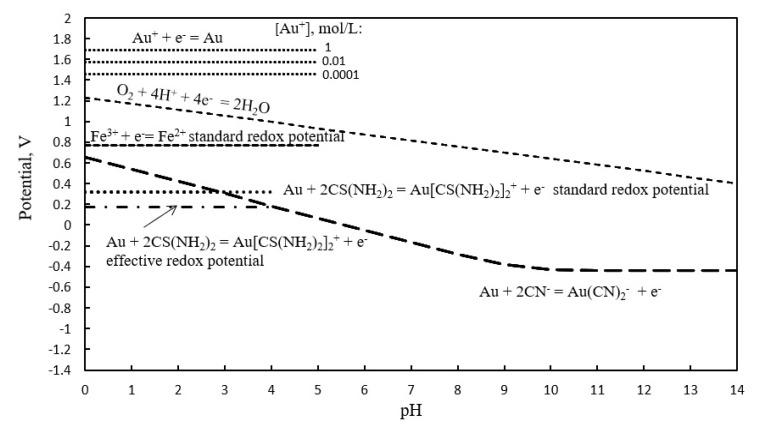
Potential-pH equilibrium diagram illustrating features of the Au-thiourea-H_2_O and Au-cyanide-H_2_O systems at 25 °C. [CN^−^]_total_ = 1 × 10^−4^ mol/L, Au(CN)_2_^−^ = 1 × 10^−3^ mol/L, Au^+^ = 5 × 10^−4^ mol/L, CS(NH_2_)_2_ = 0.33 mol/L, Fe ^3+^ = 0.11 mol/L.

**Figure 2 materials-14-00362-f002:**
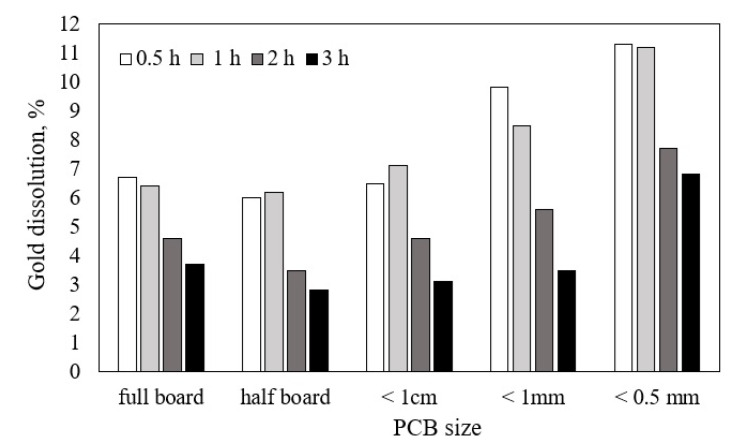
Leaching of PCB as a function of size and leach time.

**Figure 3 materials-14-00362-f003:**
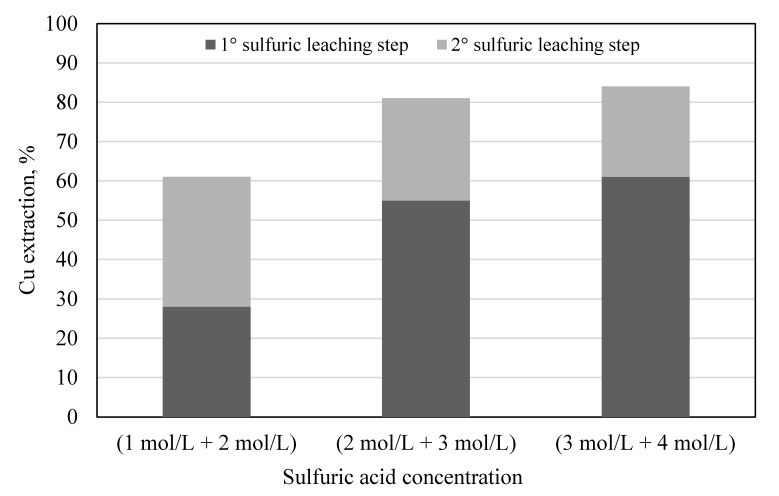
Copper dissolution yields as a function of sulfuric acid concentration in the two-step leaching. H_2_O_2_ 20% *v*/*v*, L/S ratio 10, T 25 °C, leach time 1 h, particle size −0.5 mm, and stirring 250 rpm.

**Figure 4 materials-14-00362-f004:**
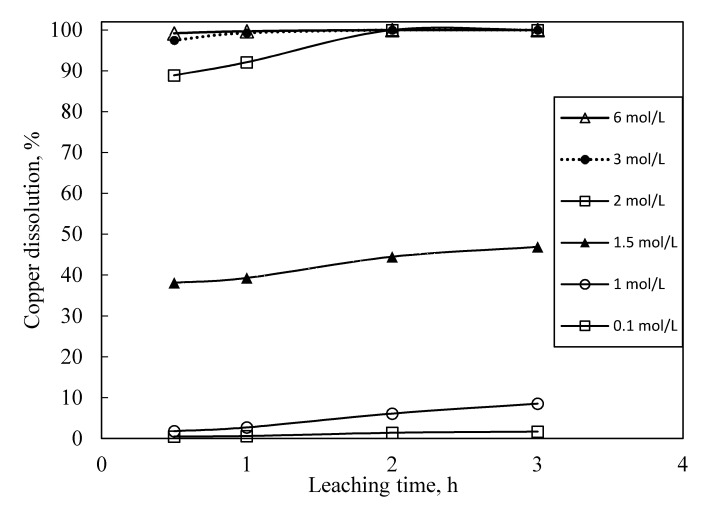
Copper dissolution yields as a function of nitric acid concentration and leach time. H_2_O_2_ 20% *v/v*, L/S ratio 10, T 25 °C, particle size −0.5 mm, and stirring 250 rpm.

**Figure 5 materials-14-00362-f005:**
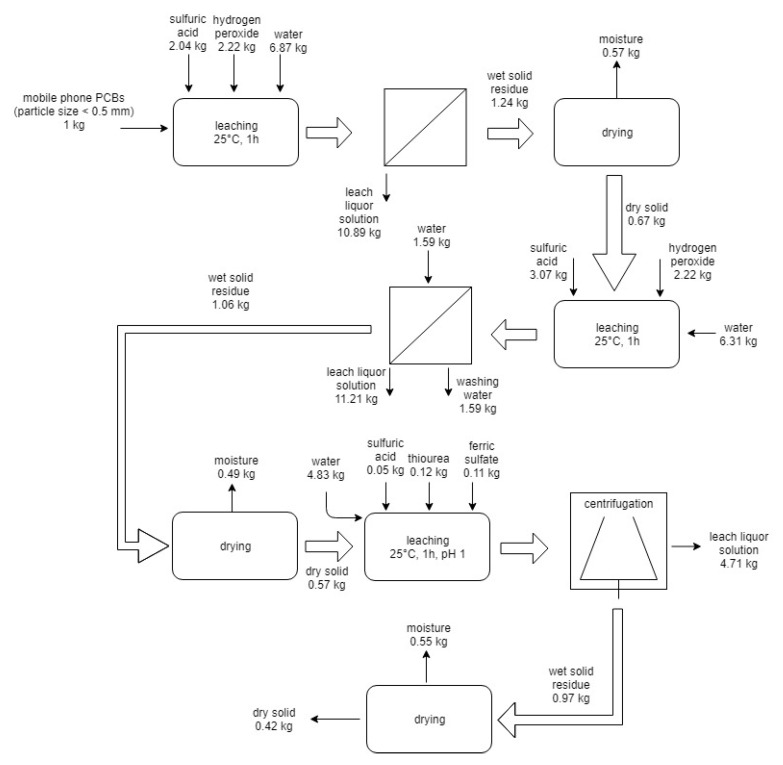
Flowsheet and mass balance of the first route process for the treatment of 1 kg of end-of-life mobile PCBs.

**Figure 6 materials-14-00362-f006:**
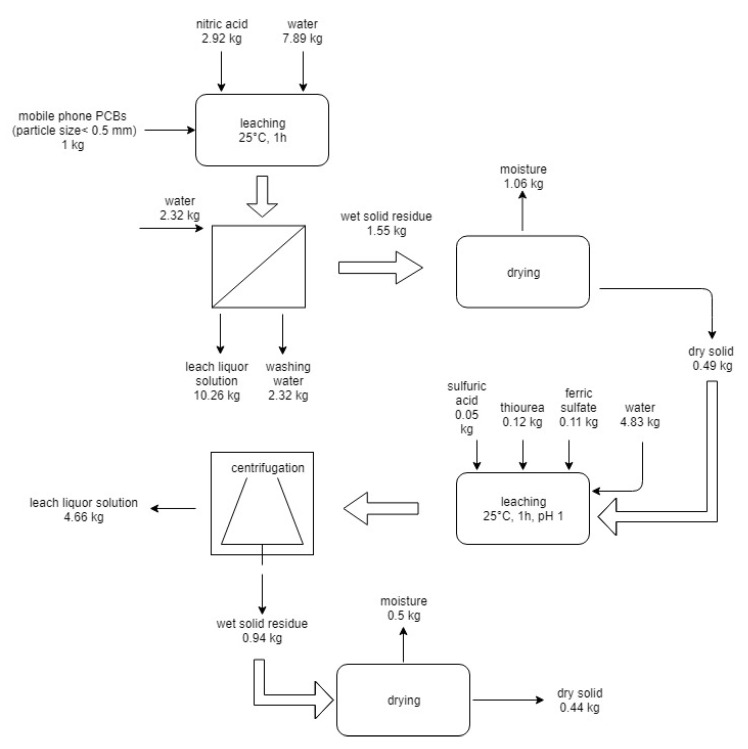
Flowsheet and mass balance of the second route process for the treatment of 1 kg of end-of-life mobile PCBs.

**Table 1 materials-14-00362-t001:** Equilibrium concentration calculated.

	Au^+^	+ 2CS(NH_2_)_2_	⇆ Au(CS(NH_2_)_2_)_2_^+^
initial, mol/L	5 × 10^−4^	0.33	-
reacted, mol/L	X	2X	-
equilibrium, mol/L	5 × 10^−4^ − X	(0.33 − 2X)	X

**Table 2 materials-14-00362-t002:** Factors and levels investigated with the full factorial plan 2^3^.

Factor	Level (−1)	Level (0)	Level (+1)
A, thiourea (g/L)	10	25	40
B, Fe^3+^ (% *w*/*v*)	0.4	0.6	0.8
C, temperature (°C)	25	52.5	80

**Table 3 materials-14-00362-t003:** Factors and levels investigated with the full factorial plan 2^2^.

Factor	Level (−1)	Level (0)	Level (+1)
A, thiourea (g/L)	40	60	80
B, L/S ratio (*v*/*w*)	5	10	15

**Table 4 materials-14-00362-t004:** Chemical composition of PCBs mobile phones (−0.5 mm) determined by atomic absorption spectroscopy (AAS). Averages of six replicates.

**Element**	**Concentration, %**	**Std Deviation, %**
Cu	27.0	1.2
Si	9.66	0.48
Fe	3.22	0.37
Sn	2.85	0.23
Al	1.98	0.21
Ni	1.45	0.24
Zn	0.59	0.11
Cr	0.13	0.05
**Precious metals**	**Concentration, mg/kg**	**Std Deviation, mg/kg**
Au	439	18
Ag	336	15
Pd	18	2

**Table 5 materials-14-00362-t005:** Gold dissolution as a function of time with direct thiourea leaching (no preliminary treatment) and after preliminary treatment with H_2_SO_4_/H_2_O_2_.

Time, h	Gold Dissolution, %	
	Direct Thiourea Leaching	Thiourea Leaching after Sulfuric Acid—Hydrogen Peroxide Treatment
0.5	9.1	79
1.0	8.5	79
2.0	7.7	78

**Table 6 materials-14-00362-t006:** 2^3^ full factorial design with replicated central point (A: thiourea, g/L; B: Fe^3+^, % *w*/*v*; C: temperature, °C).

Test	Treatment	A	B	C	Gold Dissolution, %
1	(1)	10	0.4	25	73.7
2	a	40	0.4	25	79.6
3	b	10	0.8	25	68.6
4	ab	40	0.8	25	78.4
5	c	10	0.4	80	71.9
6	ac	40	0.4	80	80.2
7	bc	10	0.8	80	68.2
8	abc	40	0.8	80	81.8
9	I	25	0.6	52.5	73.5
10	II	25	0.6	52.5	78.8
11	III	25	0.6	52.5	77.4

**Table 7 materials-14-00362-t007:** ANOVA full factorial design (A: thiourea, g/L; B: Fe^3+^, % *w*/*v*; C: temperature, °C).

Effects	Coefficient Value	Sum Square	F Value	Significance, %
A	9.40	176.62	23.43	96
B	−2.10	8.82	1.17	61
AB	2.30	10.58	1.40	64
C	0.45	0.41	0.05	16
AC	1.55	4.80	0.64	49
BC	1.05	2.20	0.29	36
ABC	0.35	0.24	0.03	13

**Table 8 materials-14-00362-t008:** Results of 2^2^ full factorial design with two replicated central points.

Treatment	Factor A, Thiourea (g/L)	Factor B, Liquid/Solid	Au Dissolution (%)
(1)	40	5	72.8
a	80	5	76.5
b	40	15	80.8
ab	80	15	84.5
I	60	10	85.8
II	60	10	84.6

**Table 9 materials-14-00362-t009:** Technical and economic feasibility of the investigated treatments as a function of the recovery of metals (X: low impact; XX: medium impact; XXX: high impact).

	Direct Leaching Thiourea	Thiourea Leaching after Copper and Base Metals Leaching (H_2_SO_4_/H_2_O_2_)	Thiourea Leaching after Copper and Base Metals Leaching (HNO_3_)
Au dissolution, %	11	79	81
Cu dissolution, %	8	81	100
Technical and Economic Feasibility			
Emissions/wastewater	X	XX	XXX
Plant complexity	X	XXX	XXX
Number of chemicals	X	XXX	XX
Costs	X	XXX	XX

## Data Availability

Data presented are openly available.

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
