# Peer review of "Effect of Acid Leaching Pre-Treatment on Gold Extraction from Printed Circuit Boards of Spent Mobile Phones"

_materials, 2021, doi:10.3390/ma14020362_

Round 1
Reviewer 1 Report
The authors have investigated two-step leaching that aiming to firstly dissolve copper in mineral acids solution, while gold dissolution in the later step with acidic thiourea solution. The topic is well within the scope of this journal and investigated a timely topic to the sustainable treatment of the increasing volume of electronic waste (herein, spent mobile phone printed circuit boards). Gold being the most precious metal therein though requires separate handling due to its low weight fraction in the PCBs. The authors have tried to well connect the experimental results with the solution chemistry of gold. However, there are several points that must be addressed before its consideration for publication, given below.
Comments to the authors:
- The authors should clarify why they have chosen H2SO4+H2O2 and HNO3 as a pre-treatment. Why HCl was not employed?
- How the authors will justify the hazardous and toxic emissions of HNO3 over other mineral acids? Certainly, the process cannot be taken as an environmentally-friendly and to solve one problem it creating another problem to the environment and surroundings. How it can be justified?
- There is no final conclusion on the use of a particular route, either H2SO4+H2O2 or HNO3? Which route can be used and why?
- Table 7 is quite confusing. How thiourea leaching after pretreatment can dissolve 81% and 100% copper? Actually, the reviewer has concerns about naming acid treatment step as pretreatment step and should be named as copper(base) metals leaching step. Also, what types of factors involving plant complexity and the number of chemicals is quite unclear.
- In the flow-sheets of mass balance, the authors should indicate the mismatch% in the balance.
- In the figures, the first letter of the heading of the axis should be capital. Also, in the tables, the first letter for labeling each column should be a capital letter.
- The authors should discuss the issue that why gold dissolution was improved while copper was pre-dissolved in the HNO3 solution.
- The authors should provide a full analysis of the waste PCBs and leach liquors as well.
- The authors should refer to and cite the following meaningful articles and book, as follows:
- J Chem Technol Biotechnol (2020), vol. 95, 2796–2810
- Renew Sustain Energy Rev. (2017), vol. 78, 220–232
- Gold Metallurgy and the Environment, (Eds.) S. Ilyas, J.-c. Lee, Boca Raton: CRC Press
- Waste Management (2021), https://doi.org/10.1016/j.wasman.2020.12.013
Author Response
Rev. 1 moderate English correction
Comments to the authors:
- The authors should clarify why they have chosen H2SO4+H2O2 and HNO3 as a pre-treatment. Why HCl was not employed?
Response 1:
The aim of the authors was to identify the best leaching system to remove copper from PCBs of spent mobile phones. The efficiency of HCl was not investigated because it would simultaneously dissolve both copper and gold, since it is a complexing agent. In order to better recover gold from the solution it is easier to dissolve gold in a different solution from copper.
The first step of leaching consisted of removal of copper that was mostly present as elemental copper (oxidation number zero). For this reason, we needed oxidizing acids like sulphuric acid aimed with H2O2 and nitric acid but not HCl that is a complexing but not oxidizing acid. Preliminary tests confirmed that no quantitative dissolution of copper was obtained with the use of HCl.
- How the authors will justify the hazardous and toxic emissions of HNO3 over other mineral acids? Certainly, the process cannot be taken as an environmentally-friendly and to solve one problem it creating another problem to the environment and surroundings. How it can be justified?
- There is no final conclusion on the use of a particular route, either H2SO4+H2O2 or HNO3? Which route can be used and why?
Response 2-3:
All the gases and vapors released during the treatment, should be captured and conveyed to an absorber plant before they are released in atmosphere. Even though, the environmental problems are not completely solved, the author think that the use of thiourea is more acceptable than the use of sodium cyanide to recover gold. The copper present in the board has to be removed with diluted acids to permit the leaching of gold in the residue left after leaching of copper.
In the section 5.1 the authors reported a discussion concerning the different investigated routes. In detail, the different treatments were evaluated from a technical, economic and environmental point of view. For the environmental impact were took into account the emissions and the production of wastewater; the HNO3 pre-treatment leaching was the treatment with the highest environmental impact (Table 7).
Hence, both the H2SO4/H2O2 and HNO3 leaching, were suitable by a technical point, but the suggested route was that with H2SO4/H2O2 because produces less pollutants emissions and allows a higher efficiency of copper recovery from the solution by adopting the electrowinning. In fact, with the HNO3 leaching the copper can be recovered by cementation processes with a lower recovery.
For these reasons the H2SO4/H2O2 route is included in the proposed recycling process.
- Table 7 is quite confusing. How thiourea leaching after pretreatment can dissolve 81% and 100% copper? Actually, the reviewer has concerns about naming acid treatment step as pretreatment step and should be named as copper(base) metals leaching step. Also, what types of factors involving plant complexity and the number of chemicals is quite unclear.
Response 4:
The authors thank the reviewer for the valuable consideration, the treatments in Table 7 were now named as suggested.
For plant complexity the authors considered the complexity of the process that depends on the number of stages, the number of chemicals and on the type of scrubber as a function of toxic emissions; based on these factors the number and the type of the equipment of the plant was estimated. The number of chemicals, for a first preliminary economic estimate, allowed to evaluate a comparison of operating costs among the proposed routes.
- In the flow-sheets of mass balance, the authors should indicate the mismatch% in the balance.
Response 5:
The authors carefully checked again the mass balances of both flowsheets and did not find any mismatch% in the balance. The sum of the inputs is equal to the sum of the outputs as it is imposed by the mass balance.
- In the figures, the first letter of the heading of the axis should be capital. Also, in the tables, the first letter for labeling each column should be a capital letter.
Response 6:
The modifications were made according to reviewer comments.
- The authors should discuss the issue that why gold dissolution was improved while copper was pre-dissolved in the HNO3 solution.
Response 7:
A slight increase in gold dissolution after thiourea leaching was achieved when pre-treatment with HNO3 was carried out. HNO3 acid leaching allowed to remove almost all the copper from PCBs powders compared to 81 % of copper dissolution that is obtained when H2SO4/H2O2 leaching was adopted. The less amount of copper left in the residue makes the dissolution of gold easier since otherwise copper would have consumed thiourea by reducing the formation of gold-thiourea complex.
Nevertheless, gold dissolutions were similar for both routes, probably because not leached copper after H2SO4/H2O2 treatment is present as “refractory material” and did not compete with gold in the successive thiourea leaching. This last statement was added in the discussion section.
- The authors should provide a full analysis of the waste PCBs and leach liquors as well.
Response 8:
The waste PCBs characterization was reported in Table 3: the concentration of precious and main metals was determined. The other fractions are constituted by plastics, ceramics and silicon wafer.
Concerning the leach liquors, in this phase of research in order to evaluate the economic and technical feasibility of the process only gold was measured. In fact, the economic value of waste PCBs is constituted for more than 90 % by gold. For preliminary leaching also copper was measured because its removal was necessary to avoid that it would competes with gold for thiourea consumption in the successive leaching.
- The authors should refer to and cite the following meaningful articles and book, as follows:
- J Chem Technol Biotechnol (2020), vol. 95, 2796–2810
- Renew Sustain Energy Rev. (2017), vol. 78, 220–232
- Gold Metallurgy and the Environment, (Eds.) S. Ilyas, J.-c. Lee, Boca Raton: CRC Press
- Waste Management (2021), https://doi.org/10.1016/j.wasman.2020.12.013
Response 9:
The suggested articles and book were added to the references of the manuscript.

Reviewer 2 Report
This manuscript investigates acid leaching (H2SO4+H2O2 / HNO3), followed by thiourea leaching with the aim of improving gold recovery from PCBs of spent mobile phones. Although the topic of this paper is of importance for "Circular Economy", there are publications of which contents are almost the same as this work.
-Birloaga et al., 2013. Waste Management 33(4), 935-941.
-Zhang et al., 2012. Procedia Environmental Sciences 16, 560-568.
Due to the lack of novelty, I think this manuscript is not enough to be published in Materials. I encourage the authors to find something new for this study.
Author Response
Rev. 2 moderate English correction
This manuscript investigates acid leaching (H2SO4+H2O2 / HNO3), followed by thiourea leaching with the aim of improving gold recovery from PCBs of spent mobile phones. Although the topic of this paper is of importance for "Circular Economy", there are publications of which contents are almost the same as this work.
-Birloaga et al., 2013. Waste Management 33(4), 935-941.
-Zhang et al., 2012. Procedia Environmental Sciences 16, 560-568.
Due to the lack of novelty, I think this manuscript is not enough to be published in Materials. I encourage the authors to find something new for this study.
Response:
The purpose of this manuscript is to identify the best pre-treatment acid leaching in order to remove copper and base metals from PCBs, since preliminary tests showed direct thiourea leaching does not allow to dissolve gold because copper inhibits the formation of gold-thiourea complex.
Moreover, with respect to the mentioned publications, in this paper a comparison among H2SO4/H2O2 and HNO3 preliminary leaching was made by a technical, economic and environmental point of view. Experimental tests showed the technical suitability of both treatments, then the authors in the discussion section also compared the economic and environmental feasibility to evaluate the best hydrometallurgical process.
In addition, the novelty of the present paper is the application of hydrometallurgical treatments on a different material (waste PCBs of spent mobile phones) with respect to the mentioned papers and to the most of scientific literature.

Reviewer 3 Report
The discussion section might be improved.
Figure 1: OY Axis description: should be “Potential, V” and values separated with dots, not comas.
Tables 1-7: table titles should be at the top, including capitalized headers
Table 1: line 2: "3+" -superscript
Figure 3: OX axis: Sulfuric (capitalized S)
Figure 4: I recommend presenting those data in the form of a table or just in the text.
Figure 5: Chart is very confusing. It would be better to place a legend in one place, e.g., at aside. Error bars should be added; the first letter of the axis name should be capitalized.
Author Response
Rev. 3 No correction English
The discussion section might be improved.
Response 1:
The authors improved the discussion section as suggested by the reviewer.
The following statement was added in the text:
lines 344-346: “Despite, different copper recoveries obtained by the two routes, gold dissolutions after thiourea leaching were similar, probably because the refractory copper not leached with the H2SO4/H2O2 preliminary treatment did not compete with gold for thiourea consumption”
Figure 1: OY Axis description: should be “Potential, V” and values separated with dots, not comas.
Response 2:
The Figure 1 was modified according to reviewer’s comments.
Tables 1-7: table titles should be at the top, including capitalized headers
Response 3:
The table titles have been moved on the top.
Table 1: line 2: "3+" -superscript
Response 4:
The correction was made.
Figure 3: OX axis: Sulfuric (capitalized S)
Response 5:
The correction was made.
Figure 4: I recommend presenting those data in the form of a table or just in the text.
Response 5:
The data now are presented in the form of a table.
Figure 5: Chart is very confusing. It would be better to place a legend in one place, e.g., at aside. Error bars should be added; the first letter of the axis name should be capitalized.
Response 5:
The authors modified the figure according to suggested comments. In the text was added the value of the experimental error (1-2 %), the error bars were not showed

Round 2
Reviewer 1 Report
The authors have significantly revised the manuscript reaching the acceptance level.
Author Response
Thanks for your revision and comments, I send you the final version of our work.

Reviewer 2 Report
I read the authors' response, but I'm not convinced by the authors, so my decision is the same as before (reject).
Author Response
Answer to the second reviewer:
I want to thank the 2nd reviewer that allows me to clarify the novelties of our work with respect to the two papers cited by him. These have been included in our paper (citation 17 and 20) and are reported below for discussion.
1)-Birloaga et al., 2013. Waste Management 33(4), 935-941.
2)-Zhang et al., 2012. Procedia Environmental Sciences 16, 560-56
1)
- a) As far as the first paper is considered, the authors treated printed circuit board from computers. In our paper only boards coming from spent mobiles were considered whose behaviors could be different from the first ones. So, the experimentation should be confirmed when a new material is involved.
This has been added in lines 68-70:
“Birloaga et al. [20] reports the leaching of spent computer printed circuit board that are different from the boards of spent mobile phones that are the material used in this paper.”
- b) Moreover, the authors carried out the oxidative leaching with H2SO4/H2O2 In our paper the two most commonly used chemical pretreatments, namely H2SO4/H2O2 and HNO3 were used on the same material and the results compared with each other.
This has been added in lines 73-75:
“…..to highlight the differences occurring on the successive gold extraction when the two different oxidant solutions are preliminarily applied on the same material.”
- c) The experimental tests were carried out with the use of factorial plan than gives a more objective judgment about the influence of the experimental parameters with respect to the purely experimental error.
This has been added in lines 78-79:
“This also permitted to determine the influence of those experimental parameters on the gold extraction against the purely experimental error”
- d) A thermodynamic diagram potential/pH was tailored on the basis of the real gold content in the boards leached in this paper. This allowed to dose the amount of reagents (thiourea and ferric sulphate) to prepare a leaching solutions having the right potential to oxidize gold. No paper dealing with thiourea has reported a tailored diagram like that we built for this particular material. (lines 142-149)
2)
- a) The paper is only a state of art of the leaching of gold from generic waste circuit boards. It reports short summaries of a good number of papers dealing with the topic. The only cited paper similar to our one, is the following:
Xu XL, Li JY. Experimental study of thiourea leaching gold and silver from waste printed circuit boards. Journal of Qingdao University (E&T), 2011, 26:69-73.
The effects of the temperature and of the liquid/solid ratio have been not considered in this paper whereas they have been tested in our paper. Moreover, the full reading of the paper is not available in the scientific literature because is written in Chinese.

Reviewer 3 Report
All changes have been accepted.
Author Response
Thanks for your review and your comments, I send you the final version of our work.
